# Genetic tracking of Crimean-Congo haemorrhagic orthonairovirus in Hyalomma population infesting cattle in Nigeria

Oluwafemi Babatunde Daodu[1]*, Joseph Ojonugwa Shaibu[2], Rosemary Ajuma Audu[2], Daniel Oladimeji Oluwayelu[3,4]*

1 Faculty of Veterinary Medicine, Department of Veterinary Microbiology, Virology Unit, University of Ilorin, Ilorin, Nigeria, 2 Nigerian Institute of Medical Research, Centre for Human Virology and Genomics, Yaba, Nigeria, 3 Faculty of Veterinary Medicine, Department of Veterinary Microbiology, Arbovirology Unit, University of Ibadan, Ibadan, Nigeria, 4 Centre for Control and Prevention of Zoonoses, University of Ibadan, Ibadan, Nigeria

* daodu.ob@unilorin.edu.ng (OBD); ogloryus@yahoo.com (DOO)

**Data Availability Statement:** All sequence files from this study are available from the NCBI database. Accession number PP645434: https://www.ncbi.nlm.nih.gov/nuccore/PP645434

## Abstract

Crimean-Congo haemorrhagic fever virus (CCHFV), a Biosafety level 4 pathogen transmitted by ticks, causes severe haemorrhagic diseases in humans but remains clinically silent in animals. Over the past forty years, Nigeria lacks comprehensive genetic data on CCHFV in livestock and ticks. This study aimed to identify and characterize CCHFV strains in cattle and their Hyalomma ticks, the primary vector, in Kwara State, Nigeria. Blood samples and Hyalomma ticks were collected from cattle, with ticks identified to species, pooled, and homogenized for RNA extraction. The CCHFV S-segment was detected using specific primers via reverse transcriptase polymerase chain reaction, followed by sequencing of amplicons. Among 318 cattle, 318 sera samples and 2855 Hyalomma ticks (*H. dromedarii* (49.0%), *H. truncatum* (44.5%), and *H. rufipes* (6.5%)) were obtained. Only two tick pools of *H. truncatum* tested (2/319 pools) were positive for CCHFV, with no positive cattle sera detected. The sequenced positive pools, denoted as CCHFV/NGR/ILN/2021/F22_S-segment (1228 bp) and CCHFV/NGR/ILN/2021/F101_S-segment (863 bp), showed 98.21% nucleotide identity with 15 variations. These strains shared 98.13% and 98.93% nucleotide identity with CCHFV IbAr10200/UCCR4401 isolated from Nigerian ticks, but only 93.88% and 93.63% similarity with CCHFV isolated in 2016 from humans in Nigeria. Additionally, compared to CCHFV isolate IbAr10200 (KY484036), sequences from this study exhibited 9–23 nucleotide variable positions with 3–4 non-synonymous amino acid replacements. Phylogenetic analysis revealed clustering of these strains around IbAr10200, suggesting ongoing circulation. This study underscores the need for broader surveillance to understand the full spectrum of CCHFV strains and clades circulating in Nigeria.

Accession number PP645435: https://www.ncbi.nlm.nih.gov/nuccore/PP645435

**Funding:** OBD received funding for this study from the Nigeria Institute of Medical Research (1NIMREX0004-19-01) for financial support of this work. The funders had no role in study design, data collection and analysis, decision to publish, or preparation of the manuscript. https://nimr.gov.ng/

**Competing interests:** The authors have declared that no competing interests exist.

# Introduction

Crimean-Congo haemorrhagic orthonairovirus (CCHFV) is the most popular tick-borne haemorrhagic virus classified among viral haemorrhagic fever (VHF) agents among humans in the world [1,2]. Unlike most VHF agents (Lassa virus, Ebola virus, Marburg virus, Rift Valley fever virus, and Yellow fever virus), people at risk of CCHFV are those who work in close contact with livestock. The virus belongs to the family *Nairoviridae* and the genus *Orthonairovius*. It is made of a negative sense single-stranded tripartite RNA genome with a bilipid envelope. Their three segmented genomes are denoted S- small segment ($\sim$1.6kb), M- medium segment ($\sim$5.4kb), and L- large segment ($\sim$12.1kb) which encode nucleoprotein, glycoproteins, and RNA-dependent RNA polymerase respectively [3]. The S-segment is the most conserved of all the segments. Presently, seven CCHFV clades have been identified in the world based on genotypic and antigenic variations, with three in Africa (Africa clades 1–3), two in Europe (Europe clades 1–2) and two in Asia (Asian clades 1–2) [4].

*Hyalomma* species has been described as the principal vector and reservoir of CCHFV [4]. Additionally, CCHFV has been isolated from *Boophilus*, *Amblyomma*, *Rhipicephalus*, *Dermacentor*, and *Culicoides* [5,6]. The principal role of *Hyalomma* ticks in the CCHFV epidemiology is further typified with evidence of parallel association of Hyalomma's geographical distribution and the CCHFV cases distribution in Africa, Southern and Eastern Europe, the Middle East, India, and Asia [4,7]. CCHFV is maintained in Hyalomma tick stages through horizontal (trans-stadial, veneral, and co-feeding) and vertical (trans-ovarian) transmission routes, and eventually transmitted to domestic animals and or humans [4,8–10]. As a two-host tick, the larval and nymphal stages of *Hyalomma* infest small mammals like giant rats, squirrels, and hares thereby maintaining CCHFV in the sylvan cycle. The adult stage infests livestock especially large mammals thereby maintaining the domestic cycle of CCHFV [8] and this seems to be the popular CCHFV source to humans.

CCHFV causes subclinical disease in most animals infected but elicit mild to severe haemorrhagic derangements in humans and may lead to death. The first human CCHFV case was reported in Asia continent among 200 Soviet military personnel in the Crimean Peninsula (1944) and in African continent in a boy who lived in the Democratic Republic of Congo (1969) [11–13]. In Nigeria, Causey et al [14] reported the first CCHFV isolation from ticks collected between 1964–1968 which was later characterized. Most studies in Nigeria were mainly based on antibody detections in humans and animals [15–18] not until 2016 when the first human case was confirmed (Borno strain) in a female febrile patient [19].

Although an unpublished CCHFV sequence from Nigerian tick (accession number KY484036) was submitted to the National Center for Biotechnology Information in 1996, there exist scanty information about CCHFV strain/clade circulating among this important competent vector in Nigeria. Hence, a need to update the CCHFV record for adequate disease preparedness. Moreso, the nucleotide divergence between the first CCHFV isolate (IbAr10200 strain; Year: 1968; Source: tick) and the CCHFV Borno strain (Year: 2016; Source: Human) suggest possible variations among CCHFV strains circulating in Nigeria with different virulence. This study was designed to know CCHFV circulating in the cattle population and their infesting Hyalomma ticks in Kwara State Nigeria. It further investigated possible genetic mutations in circulating virus.

## Materials and methods

### Ethical approval

Approval was obtained from ethical review committees of the Kwara State Ministry of Agriculture and Rural Development (VKW-714/I/83) and the Animal Care and Use Research Ethics Committee of the University of Ibadan (UI-ACUREC/092-1121/18). The informed consent of animal owners was sort verbally before sampling.

### Study location, sampling design, and duration

The study was conducted in a major abattoir (Abubakar Bukola Saraki Memorial modern abattoir) in Kwara State of Nigeria. The State is a major transit state for ruminant transportation from Northern Nigeria to the South-western part (Fig 1). The abattoir slaughters an average of 100 cattle/day excluding Sunday when no activity takes place. Based on empirical evidence, the cattle slaughtered in this abattoir originate from Kwara, Niger and Oyo States as well as some States in the Northern part of Nigeria, where they were reared often under extensive and sometimes semi-intensive management system. The previous study indicated that cattle slaughtered in this abattoir had a high CCHFV IgG prevalence as compared with other abattoirs in the State [20]. Slaughtering and carcass processing occur at the slab (8 partitions) of the abattoir. The abattoir has an estimated 540 abattoir workers (full-time and part-time) excluding people who sell foodstuffs and other non-animal related goods. Cattle are not often examined and prepared (de-tick, washed) before passing for slaughter. The timeframe from cattle entry to slaughter at the abattoir is often <24 hours. For this study, samples were obtained during the dry seasons (December–March) which spanned between 2018–2019 and 2019–2020. Dry season was selected based on previous pilot study conducted which indicated the abundance of the Hyalomma tick infestation on cattle during this time. Higher CCHFV seroprevalence was also reported among cattle during this season in the same study location as compared with the rainy season [20].

A cross-sectional design was adopted with random sampling technique. Also, a sample size of 281 was calculated using a formular by Thrustfield [21] (Sample size = $z^2$ p (1-p)/$d^2$, where z = confidence interval at 95%, p = disease prevalence (24% based on Oluwayelu et al [18]),

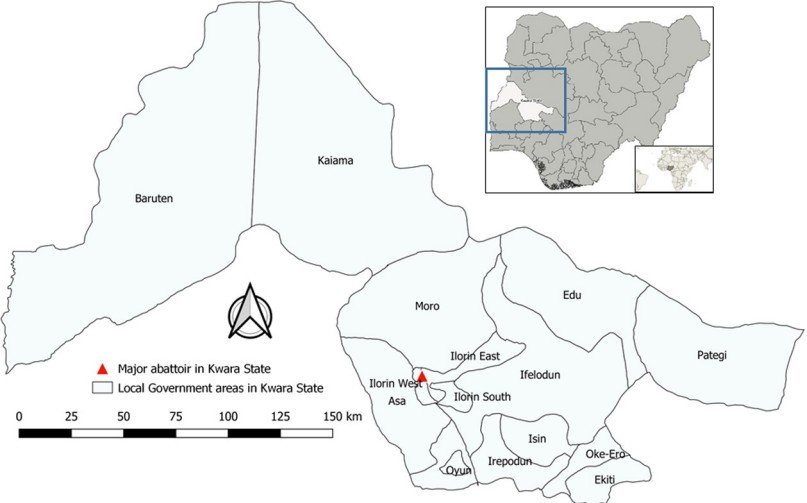

**Fig 1. Map of Kwara State indicating (red colour triangle) the major abattoir in the State.** The base layer of the map was created using DIVA-GIS Version 7.5 (https://www.diva-gis.org/) software.

d = precision value at 5%). In order to account for attrition, 13% of the estimated sample size was added making a total size of 318 cattle.

## Blood collection

Five (5) milliliters of blood was obtained aseptically from cattle through the jugular vein (ensuring minimal pain with application of methylated spirit for disinfection) and was dispensed into a plain tube. The tube was slanted into a cold pack and transported to the laboratory. The serum was harvested from the clotted blood after centrifugation at 4000 revolutions per minute, and stored at -20°C until use for CCHFV molecular detection.

## Tick collection and processing

Hyalomma ticks from the cattle bled (2–15 Hyalomma ticks/cattle) were obtained and transported under the cold chain to the laboratory for species identification and molecular investigation. Hyalomma ticks from each cattle were washed briskly in sterile water (twice) to remove dirt and then identified to the species level using established morphological keys [22,23]. Subsequently, ticks were then made into groups of 2 to 15 ticks/pool (based on the date of collection, individual cattle, and tick species), homogenized in RNA later, and spanned at 10,000 rpm for 10 minutes. The supernatants were then harvested and prepared for RNA extraction.

## RNA extraction and primer design

The viral nucleic acid was extracted from sera and tick homogenates, using the viral RNA +DNA prep. kit (Jena Bioscience, GmBH), according to the manufacturer's instructions. Primers were designed in-house from conserved regions using Oligo Primer Analysis Software v.7 to obtain amplicons of 974 – 1361bp [24]. The primers were synthesized by Macrogen Europe B.V. (Amsterdam, Netherlands). The primers (CCHFV_NG_F1-TTACCTTCTGAGTGTTAG CAAAATGGand CCHFV_NG_R1- ATGCCATGTCCTGTATGTTGAATCC) amplify nucleotide (nt) region 36nt to 1396nt of the CCHFV S segment giving an amplicon length 1361bp [24]. Additionally, two nested primer pairs were designed (CCHFV_NG_F2-TGAGTGGTTCGA TTAAAAAATGCAGG, CCHFV_ NG_R2- TCCGAAGGCTTGAGAATGGACTTGG, CCHFV_ NG_F3-CCTATTCCTTATTGCGAGAGTGTTCCand CCHFV_ NG_R3- AGAGGCCACTTATGT CCATGTCC).

## Nucleic acid amplification

Complementary DNA was synthesized, using the SCRIPT cDNA Synthesis kit (Jena Bioscience, GmBH), according to manufacturer's instructions. Subsequently, a gradient polymerase chain reaction (PCR)was conducted with each primer pair to optimize the annealing temperature and amplification of the targeted fragment. This was carried out using MiniAmp Plus Thermal Cycler (Applied Biosystems). The PCR, involving a total volume of 20μl, included 5μl of 5x Red Load Taq Master mix, 0.4μl each primer, 5μl cDNA, and 9.2μl nuclease-free water. The following thermal cycle conditions were used: 94°C for 2 minutes, 30 cycles of 94°C for 30 secs, 57°C for 30 secs and 72°C for 2 minutes, and a final extension at 72°C for 2 minutes. The amplicons were analyzed in 1.8% agarose gel electrophoresis.

## Sequencing and phylogenetic analysis

The amplicons were purified using HT Exo SAP-IT (Thermo Fischer Scientific) according to the manufacturer's instructions. Sequencing was then performed using BigDye Terminator kit v.3l1 and cleaned up with BigDye XTerminator v. 3.3 (Applied Biosystems, CA). Analysis of

the purified product was conducted in ABI 3130xl Genetic Analyser (Applied Biosystems). The sequence generated was subsequently assembled using DNASTAR Lasergene v. 7 and a BLAST performed in NCBI to know the related strain. Multiple alignment of the sequences was carried out, along with 30 CCHFV strains downloaded from the GenBank database of NCBI (https://www.ncbi.nlm.nih.gov/GenBank), using MAFFT software v. 7 [25]. Phylogenetic analysis was conducted with MEGA X applying the Maximum likelihood (ML) method and the Tamura-Nei model [26]. The bootstrap consensus tree was inferred from 1000 replicates based on General Time Reversible (GTR) substitution model with gamma distribution.

## Statistical analysis

Data obtained were entered into the Statistical Package for Social Sciences software, v. 22 (SPSS, USA). Descriptive statistics were analyzed.

## Result

A total of 2855 *Hyalomma* ticks were obtained from 318 cattle during the dry seasons in 2019–2020 and 2020–2021. Only three species of *Hyalomma* were seen on the cattle sampled including *H. dromedarii* (49.0%; 1398/2855), *H. truncatum* (44.5%; 1271/2855), and *H. rufipes* (6.5%; 186/2855). Also, adult male (80.3%; 2293/2855) and adult female (19.7%; 562/2855) ticks were the only tick stages seen on the cattle (Table 1). Male cattle had a higher Hyalomma tick count (9 ticks/cattle) than female cattle (4 ticks/ cattle) (Table 2). Also, cattle >5.5 years old had the highest hyalomma tick count (10 ticks/cattle). Based on the breed of cattle, the highest *H. rufipes* (six ticks/cattle), *H. dromedarii* (nine ticks/cattle), and *H. truncatum* (six ticks/cattle) were found on Friesian crossbreed, Adamawa Gudali breed and White Fulani breed of cattle respectively (Table 2).

A total of 319 tick pools were made from 2855 ticks. Only two pools (F 22 and F 101) were positive for S-segment of CCHFV (Table 3). However, none of the cattle sera tested (0/318) had detectable CCHFV S segment.

Sequences obtained were denoted as CCHFV/NGR/ILN/2021/F22_S segment (1228 bp) and CCHFV/NGR/ILN/2021/F101_S segment (863bp). These sequences had 98.21% nucleotide similarity with 15 nucleotide differences as well as 98.93% amino acid similarity with three amino acid difference. Additionally, CCHFV/NGR/ILN/2021/F22_S segment and CCHFV/NGR/ILN/2021/F101_S segment strains clustered around and had 98.13% and 98.93% nucleotide identities, respectively, with CCHFV isolate IbAr10200 (Accession number: KY484036) from Nigeria (Fig 2 and Tables 4 and 5). Both CCHFV sequences have been submitted to the GenBank (CCHFV/NGR/ILN/2021/F22: Accession number PP645434- https://www.ncbi.nlm.nih.gov/nuccore/PP645434; CCHFV/NGR/ILN/2021/F101: Accession number PP645435- https://www.ncbi.nlm.nih.gov/nuccore/PP645435).

**Table 1. Distribution and developmental stages of *Hyalomma* species infesting cattle at an abattoir in Nigeria.**

| Tick species | Adult male (%) | Adult female (%) | Total count (%) |
|---|---|---|---|
| *H. rufipes* | 157 (84.4) | 29 (15.6) | **186 (6.5)** |
| *H. dromedarii* | 1255 (89.8) | 143 (10.2) | **1398 (49.0)** |
| *H. truncatum* | 881 (69.3) | 390 (30.7) | **1271 (44.5)** |
| **Total count (%)** | **2293 (80.3)** | **562 (19.7)** | **2855** |

**Table 2. Distribution of *Hyalomma* ticks infesting cattle at an abattoir in Nigeria.**

| Feature | No of Cattle | *H. rufipes* | | *H. dromedarii* | | *H. truncatum* | | Total *Hyalomma* count | | |
|---|---|---|---|---|---|---|---|---|---|---|
| | | Mean ± SD | SE | Mean ± SD | SE | Mean ± SD | SE | Mean ± SD | SE | Total count (%) |
| Sex | | | | | | | | | | |
| Male | 4 | 0.0 ± 0.0 | 0.0 | 2.3 ± 4.5 | 2.3 | 1.3 ± 2.5 | 1.3 | 3.5 ± 4.4 | 2.2 | 14 (0.4) |
| Female | 413 | 0.6 ± 1.3 | 0.1 | 4.4 ± 5.8 | 0.3 | 4.0 ± 4.6 | 0.3 | 9.0 ± 5.2 | 0.3 | 2841 (99.6) |
| Age (year) | | | | | | | | | | |
| 1.6–2.5 | 4 | 0.8 ± 1.0 | 0.5 | 0.0 ± 0.0 | 0.0 | 6.8 ± 3.6 | 1.8 | 7.5 ± 3.0 | 1.5 | 30 (1.1) |
| 2.6–3.5 | 17 | 0.2 ± 0.5 | 0.1 | 2.1 ± 2.8 | 0.7 | 5.0 ± 3.6 | 0.9 | 7.3 ± 3.0 | 0.7 | 124 (4.3) |
| 3.6–4.5 | 109 | 0.7 ± 1.4 | 0.1 | 2.8 ± 4.5 | 0.4 | 4.4 ± 4.6 | 0.4 | 7.9 ± 4.6 | 0.4 | 859 (30.1) |
| 4.6–5.5 | 5 | 0.6 ± 0.5 | 0.2 | 2.6 ± 5.8 | 2.6 | 2.0 ± 2.8 | 1.3 | 5.2 ± 5.6 | 2.5 | 26 (0.9) |
| 5.6–6.5 | 76 | 0.8 ± 1.6 | 0.2 | 5.3 ± 6.5 | 0.7 | 3.8 ± 4.9 | 0.6 | 10.0 ± 5.5 | 0.6 | 758 (26.5) |
| >6.5 | 107 | 0.4 ± 1.0 | 0.1 | 6.0 ± 6.3 | 0.6 | 3.6 ± 4.4 | 0.4 | 9.9 ± 5.7 | 0.6 | 1058 (37.1) |
| Breed | | | | | | | | | | |
| SG | 68 | 1.0 ± 1.7 | 0.2 | 0.8 ± 2.5 | 0.3 | 5.8 ± 5.0 | 0.6 | 7.6 ± 5.2 | 0.6 | 516 (18.1) |
| AG | 127 | 0.3 ± 0.9 | 0.1 | 8.7 ± 5.8 | 0.5 | 1.7 ± 3.2 | 0.3 | 10.8 ± 5.3 | 0.5 | 1374 (48.1) |
| WF | 77 | 0.6 ± 1.3 | 0.1 | 0.3 ± 0.8 | 0.1 | 6.4 ± 4.4 | 0.5 | 7.3 ± 4.3 | 0.5 | 564 (19.8) |
| RB | 15 | 0.4 ± 0.7 | 0.2 | 4.7 ± 5.3 | 1.4 | 2.9 ± 3.6 | 0.9 | 8.0 ± 4.7 | 1.2 | 120 (4.2) |
| SG x AG | 6 | 0.0 ± 0.0 | 0.0 | 7.8 ± 6.9 | 2.8 | 2.7 ± 3.0 | 1.2 | 10.5 ± 5.3 | 2.2 | 63 (2.2) |
| WF x SG | 2 | 1.5 ± 2.1 | 1.5 | 0.0 ± 0.0 | 0.0 | 5.0 ± 7.1 | 5.0 | 6.5 ± 4.9 | 3.5 | 13 (0.5) |
| F x | 1 | 6.0 ± 0.0 | 0.0 | 0.0 ± 0.0 | 0.0 | 5.0 ± 0.0 | 0.0 | 11.0 ± 0.0 | 0.0 | 11 (0.4) |
| Others | 22 | 0.6 ± 0.9 | 0.2 | 4.3 ± 5.9 | 1.2 | 3.9 ± 4.8 | 1.0 | 8.8 ± 5.6 | 1.2 | 194 (6.8) |
| **Total** | **318** | **0.6 ± 1.3** | **0.1** | **4.4 ± 5.8** | **0.3** | **4.0 ± 4.5** | **0.3** | **9.0 ± 5.3** | **0.3** | **2855** |

Key: AG- Adamawa Gudali breed WF- White Fulani breed RB- Red Bororo breed F- Friesian breed SG- Sokoto Gudali x- cross breed SD- Standard deviation SE- Standard error of mean.

## Discussion

*Hyalomma* ticks have been identified as the principal vector of CCHFV owing to their competence in trans-stadial and trans-ovarian transmission of the virus. Additionally, the geographical distribution of *Hyalomma* species parallels the geo-spread of this virus among animals and human populations. In this study, three *Hyalomma* species were seen including *H. dromedarii* (49.0%; 1398/2855), *H. truncatum* (44.5%; 1271/2855), and *H. rufipes* (6.5%; 186/2855). These tick species had variable counts across the breeds of cattle examined. Previous reports indicated that these tick species are popular among Hyalommas infesting cattle in Nigeria [27,28]. Our analysis showed that an average count (min.) of nine *Hyalomma* ticks infest every cattle brought to the abattoir during the dry season. This indicates high tick burden and suggest a possible Hyalomma contamination of the abattoir environment with an expected increase CCHFV risk to humans working in this abattoir.

**Table 3. Features of CCHFV positive pools of Hyalomma tick infesting cattle at an abattoir in Nigeria.**

| Pool number | Number of ticks (Adult male/Adult female) | Tick species | Cattle features | | | | |
|---|---|---|---|---|---|---|---|
| | | | Breed | Sex | Age (year) | Tick infestation score | Body condition score |
| F22 | 13 (11 /2) | *H. truncatum* | WF | F | 6 | > 100 ticks | Good |
| F 101 | 10 (9/1) | *H. truncatum* | WF x SG | F | 4.5 | <50 ticks | Fair |

Key: WF- White Fulani SG- Sokoto Gudali x- cross breed F- female.

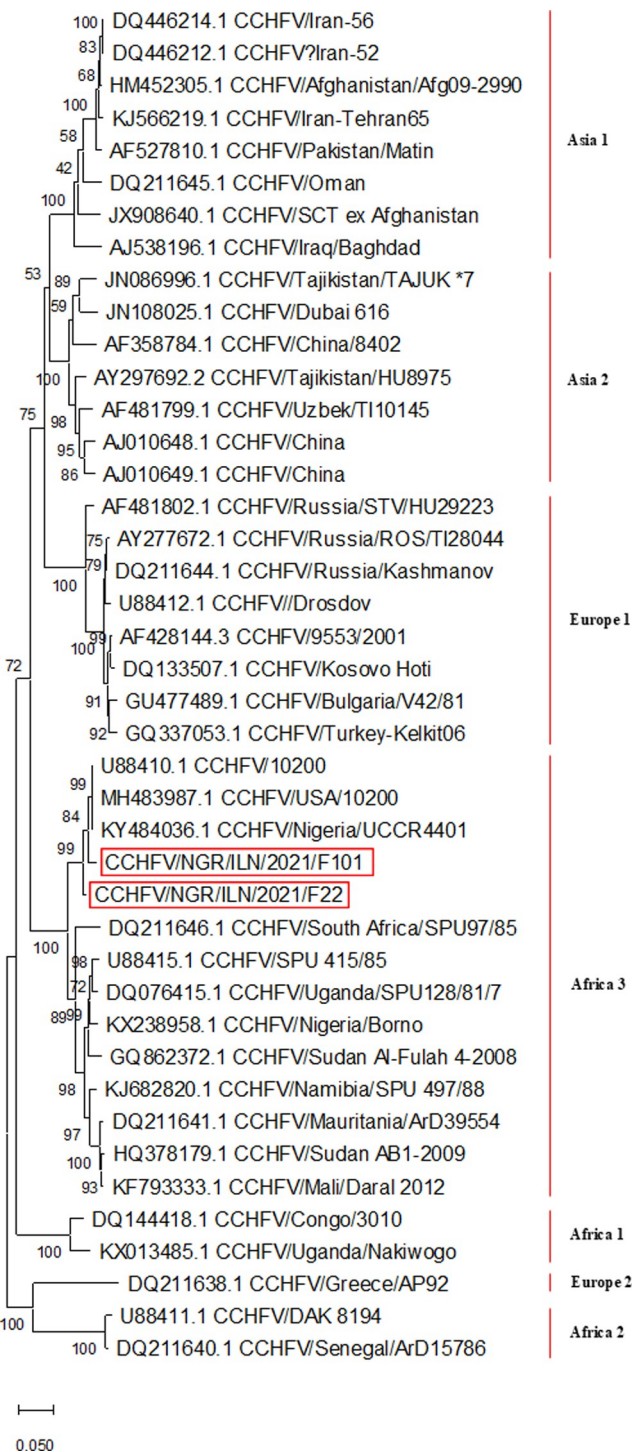

**Fig 2. Phylogenetic tree of two new CCHFV strains in relation to published strains.**

The detection of CCHFV in *Hyalomma* ticks infesting cattle in this study indicated that these ticks maintain the virus in livestock populations as well as in the environment. However, the absence of detectable CCHFV S segment in the cattle sera, especially cattle infested with CCHFV-positive *Hyalomma* ticks, suggests that either the sampling period was outside the

Table 4. Nucleotide identities of two new CCHFV strains with published strains.

| Strain/Isolate Name | Country | Accession number | Nucleotide identity (%) | |
|---|---|---|---|---|
| | | | CCHFV/NGR/ILN/2021/F22_S segment | CCHFV/NGR/ILN/2021/F101_S segment |
| IbAr10200 /UCCR4401) | Nigeria | KY484036 | 98.13 | 98.93 |
| CCHFV 10200 | USA | MH483987.1 | 98.13 | 98.93 |
| Borno | Nigeria | KX238958.1 | 93.88 | 93.63 |
| ArD39554 | Mauritania | DQ211641.1 | 93.22 | 92.69 |
| AB1-2009 | Sudan | HQ378179.1 | 93.14 | 92.45 |

viraemic phase or CCHFV was not successfully transmitted to the cattle during feeding. This further supports the reasons for reported failures in the detection of CCHFV in animal sera [17]. Additionally, it is also possible that CCHFV-positive *Hyalomma* ticks were infected by the previous host or during co-feeding with infected ticks. Although, as observed in this study, the detection of CCHFV in *Hyalomma* ticks infesting CCHFV-negative animal hosts might suggest higher risk when in contact with *Hyalomma* ticks than with cattle blood, exposure to a viraemic cattle could place several people (abattoir workers) at risk of infection within a short timeframe.

Furthermore, despite the fact that *H. dromedarii* was the most abundant *Hyalomma* species counted in this study, the two positive pools were *H. truncatum*. The reason for this could not be easily traced as CCHFV has been detected in other *Hyalomma* species. It might be that only *H. truncatum* was most abundant in the region where the cattle infested with CCHFV-borne ticks were reared. Furthermore, this study observed that the positive pools were *H. truncatum* with each pool originating from different cattle. It should note that this study was limited, in that individual ticks that made up each pool were not tested (to know the actual number of positive ticks). The detection of CCHFV positive tick suggests that the livestock farms and or pastoralist settlements these cattle originated might be infested with ticks carrying CCHFV, and the livestock workers involved are at risk of CCHFV exposure. Hence, studies investigating the spread of CCHFV in humans and animals in these localities are expedient.

Evidence of variation between the two CCHFV isolates from this study (98.21% homologous) suggests they might have originated from different sources. However, nucleotide analysis showed that they had 98.13–98.93% identity with CCHFV isolate IbAr10200 (isolated from ticks in Nigeria with accession number: KY484036) but distant (93.63–93.88% homologous) from 2016 CCHFV strain isolated from a febrile human patient admitted into a tertiary hospital in Borno State Nigeria (KX238958.1) [19]. Also, the phylogenetic tree constructed indicated that our isolates clustered around CCHFV isolate IbAr10200 but were distant from CCHFV Borno strain.

Based on comparison with CCHFV isolate IbAr10200 (Accession number: KY484036) isolated from Nigeria, CCHFV/NGR/ILN/2021/F22 had 23 nucleotides and 3 amino acid differences while CCHFV/NGR/ILN/2021/F101 had 9 nucleotides and 4 amino acid difference (Table 5).

While the amino acid types were maintained for H195R and I246V in decoded CCHFV/NGR/ILN/2021/F22 nucleotide sequence, the hydrophilic polar uncharged Serine (S) was switched to hydrophobic aliphatic Glycine (G) at amino acid position 301. Also, out of four amino acids replaced in CCHFV/NGR/ILN/2021/F101 sequence, only two mutations maintained the amino acid type (N144S and H195R) while there were switches from hydrophilic polar uncharged to hydrophobic aliphatic amino acid at amino acid position 145 (T A) and 301 (S G). These major mutations in amino acid might likely change the characteristic of the virus, hence further study is needed to properly characterise circulating CCHFV in Nigeria by

**Table 5. Mutational analysis between our sequences and CCHFV prototype strain obtained from ticks in Nigeria.**

|  | CCHFV/NGR/ILN/2021/F22 | CCHFV/NGR/ILN/2021/F101 |  |
|---|---|---|---|
| Percentage similarity (%) | 98.13 | 98.93 | CCHFV isolate IbAr10200 (Accession number: KY484036) |
| No. of Nucleotide difference | 23 | 9 |  |
| Percentage similarity (%) | 99.26 | 98.58 | CCHFV isolate IbAr10200 (Accession number: ARB51456.1) |
| No. of Amino Acid difference | 3 | 4 |  |
| Mutated region(s) | H195R, I246V and S301G | N144S, T145A, H195R and S301G |  |

whole genome sequencing to examine all possible mutations. Additionally, there is also a need to advocate for routine control of ticks in the abattoir environment to reduce their population and human contact.

## Conclusion

This study indicated that CCHFV is still circulating in Hyalomma ticks infesting cattle in Nigeria and might be of varying strains. Hence, his study underscores the need for broader surveillance to understand the full spectrum of CCHFV strains and clades circulating in Nigeria.

## Acknowledgments

The authors appreciate the cooperation of abattoir associations at the study sites.

## Author Contributions

**Conceptualization:** Oluwafemi Babatunde Daodu, Rosemary Ajuma Audu, Daniel Oladimeji Oluwayelu.

**Data curation:** Oluwafemi Babatunde Daodu.

**Formal analysis:** Oluwafemi Babatunde Daodu, Joseph Ojonugwa Shaibu.

**Funding acquisition:** Oluwafemi Babatunde Daodu, Rosemary Ajuma Audu.

**Investigation:** Oluwafemi Babatunde Daodu, Joseph Ojonugwa Shaibu.

**Methodology:** Oluwafemi Babatunde Daodu, Joseph Ojonugwa Shaibu, Daniel Oladimeji Oluwayelu.

**Project administration:** Oluwafemi Babatunde Daodu, Rosemary Ajuma Audu.

**Resources:** Oluwafemi Babatunde Daodu, Daniel Oladimeji Oluwayelu.

**Software:** Oluwafemi Babatunde Daodu, Joseph Ojonugwa Shaibu.

**Supervision:** Rosemary Ajuma Audu, Daniel Oladimeji Oluwayelu.

**Validation:** Oluwafemi Babatunde Daodu, Joseph Ojonugwa Shaibu.

**Visualization:** Oluwafemi Babatunde Daodu, Joseph Ojonugwa Shaibu.

**Writing – original draft:** Oluwafemi Babatunde Daodu.

**Writing – review & editing:** Oluwafemi Babatunde Daodu, Joseph Ojonugwa Shaibu, Rosemary Ajuma Audu, Daniel Oladimeji Oluwayelu.

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
