## [Decision Letter · Decision Letter 0]

23 Jul 2024

PONE-D-24-16845GENETIC TRACKING OF CRIMEAN-CONGO HAEMORRHAGIC ORTHONAIROVIRUS IN HYALOMMA POPULATION INFESTING CATTLE IN NIGERIAPLOS ONE

Dear Dr. DAODU,

Thank you for submitting your manuscript to PLOS ONE. After careful consideration, we feel that it has merit but does not fully meet PLOS ONE’s publication criteria as it currently stands. Therefore, we invite you to submit a revised version of the manuscript that addresses the points raised during the review process.

Please address all the comments made by reviewers. 

We look forward to receiving your revised manuscript.

Kind regards,

Engin Berber, D.V.M., Ph.D.

Academic Editor

PLOS ONE

Journal Requirements:

2. To comply with PLOS ONE submissions requirements, in your Methods section, please provide additional information regarding the experiments involving animals and ensure you have included details on (1) methods of anesthesia and/or analgesia, and (2) efforts to alleviate suffering.

3. In your Methods section, please provide additional details regarding participant consent from the owners of the animals. In the ethics statement in the Methods and online submission information, please ensure that you have specified (1) whether consent was informed and (2) what type you obtained (for instance, written or verbal). If the need for consent was waived by the ethics committee, please include this information.

Please upload the completed Content Permission Form or other proof of granted permissions as an """"Other"""" file with your submission.

Reviewers' comments:

Reviewer's Responses to Questions

**Comments to the Author**

1. Is the manuscript technically sound, and do the data support the conclusions?

Reviewer #1: Yes

Reviewer #2: Yes

2. Has the statistical analysis been performed appropriately and rigorously? 

Reviewer #1: I Don't Know

Reviewer #2: Yes

3. Have the authors made all data underlying the findings in their manuscript fully available?

Reviewer #1: Yes

Reviewer #2: Yes

4. Is the manuscript presented in an intelligible fashion and written in standard English?

Reviewer #1: Yes

Reviewer #2: Yes

5. Review Comments to the Author

Reviewer #1: Comments on the Manuscript:

The manuscript contains original data on the strain of the CCHF virus in Nigeria. However, I would like to draw attention to some methodological issues:

1. Sample Collection:

- The samples were collected from cattle brought to the slaughterhouse. However, there is no information about the location from which these cattle were brought.

- It is unclear whether all the cattle were grazing on pastures or if they were sourced from intensive farming operations. This information is crucial for understanding the epidemiological context.

2. Sampling Period:

- The authors collected samples between December and March. The rationale behind choosing this particular timeframe has not been provided. Why were samples collected specifically during these months? Are these the months when Hyalomma ticks are active or host seeking session? If so, please provide reference.

3. Tick Collection Data:

- The manuscript should include the minimum and maximum number of ticks collected from each animal.

- Additionally, it would be beneficial to provide a table detailing the number of ticks collected from each animal by month, including the species of ticks identified.

4. cDNA Synthesis Verification:

- The entire tick pools could have been checked for housekeeping genes post-cDNA synthesis. How did the authors ensure that the cDNA synthesis was accurate?

5. Specific Comments on Table 2:

- Please provide more information regarding the SG abbreveation in Table 2.

6. Serological Status of Cattle:

- Why were the cattle not found to be seropositive? How do the authors explain this finding?

7. Minor Corrections:

- Line 339: "hyalomma" should be corrected to "Hyalomma".

I hope these suggestions help improve the clarity and quality of the manuscript.

Best regards,

Reviewer #2: CCHFV is a zoonotic pathogen known to cause severe infections in humans in close contact with animals that serve as amplifying hosts and are infested with tick vectors. The study presents interesting data that adds knowledge to the distribution of the virus in Nigeria.

There are some minor changes that need to be made by the authors.

Here are some comments to consider:

Abstract:

L35: The percentages of the Hyalomma species add up to 100.4%, not 100%. Kindly check and adjust.

Introduction:

L77: Change “leaved” to “lived”.

L89: Change “hyalomma” to “Hyalomma”.

Tick collection and processing:

L133-L140: I assume all ticks present on the cattle were collected, and sent to the laboratory for identification and only the Hyalomma species processed for RNA extraction. If so, I suggest the section reflects exactly that.

Sequencing and phylogenetic analysis:

L172-L172: I recommend changing the sentence to “The sequence generated was subsequently assembled using DNASTAR Lasergene v. 7 and a BLAST performed in NCBI

to identify related strain”.

Results:

If all tick species were collected, please indicate the different species identified and narrow down to the Hyalomma species.

Again, why were other tick species such as Amblyomma variegatum not screened for CCHFV? These have been reported to carry the virus and could potentially contribute to its circulation in Nigeria.

L187: Again, the percentages of the Hyalomma species add up to 100.4%, not 100%. Kindly check and adjust.

Table 2: Kindly ensure that all the Breed is captured in the key.

L253: Please revise the sentence to be clear and easy to read.

Discussion:

L334–335: The percentages of the Hyalomma species add up to 100.4%, not 100%. Kindly check and adjust.

L337: Change “hyalomma” to “Hyalomma”.

L339: Change “hyalomma” to “Hyalomma”.

L358-359: The sentence should be revised to; “It should be noted that this study was limited, in that individual ticks that made up each pool were not tested (to know the actual number of positive ticks).

L381: Change “characterised” to “characterise”

I suggest you include some limitations of the study at the end of the discussion. By way of recommendation, you can suggest that studies sequence the whole genome of CCHFV to determine if there is reassortment in any of the segments. Reassortment in the M segment for instance, has been reported to influence the pathogenicity and epidemiology of the virus.

Reference: Burt, F. J., Paweska, J. T., Ashkettle, B., & Swanepoel, R. (2009). Genetic relationship in southern African Crimean-Congo haemorrhagic fever virus isolates: Evidence for occurrence of reassortment. Epidemiology and Infection, 137(9), 1302–1308. https://doi.org/10.1017/S0950268808001878

6. PLOS authors have the option to publish the peer review history of their article (what does this mean?). If published, this will include your full peer review and any attached files.

Reviewer #1: No

Reviewer #2: No

---

## [Author Response · Author response to Decision Letter 0]

18 Sep 2024

The only issue raised was by the editorial and responses has been uploaded as "Others"

---

## [Editor Report · Decision Letter 1]

1 Oct 2024

PONE-D-24-16845R1

GENETIC TRACKING OF CRIMEAN-CONGO HAEMORRHAGIC ORTHONAIROVIRUS IN HYALOMMA POPULATION INFESTING CATTLE IN NIGERIA

PLOS ONE

Dear Dr. DAODU,

Thank you for submitting your manuscript to PLOS ONE. After careful consideration, we have decided that your manuscript does not meet our criteria for publication and must therefore be rejected.

Specifically:

Thanks for your submission the "Revision 1". We have detected that you have failed to meet the following points. 

A rebuttal letter that responds to each point raised by the academic editor and reviewer(s). You should upload this letter as a separate file labeled 'Response to Reviewers'.You have received several comments from the reviewers, but your response was, 'The only issue raised was by the editorial, and the responses have been uploaded as "Others.". A marked-up copy of your manuscript that highlights changes made to the original version. You should upload this as a separate file labeled 'Revised Manuscript with Track Changes'.You have made several changes raised by the reviewers to the original version, but you did not mark all the changes in the 'Revised Manuscript with Track Changes.'

Therefore, your revised manuscript does not meet our criteria for publication and must therefore be rejected.

I am sorry that we cannot be more positive on this occasion, but hope that you appreciate the reasons for this decision.

Kind regards,

Engin Berber, D.V.M., Ph.D.

Academic Editor

PLOS ONE

- - - - -

---

## [Author Response · Author response to Decision Letter 1]

10 Oct 2024

RESPONSES: On 26th August 2024, I responded to all the reviews’ comments. I uploaded it and submitted and this was acknowledged. I highlighted my corrections effected in the manuscript with track changes but I only submitted responses to the reviewers’ comments. I forgot to upload my response to the Editorial comment.

Unfortunately, the manuscript was sent back to me on 27th August and 18th September and each time I was confused because the editorial comments for 27th August and 18th September were already addressed. 

I think the issue is more of misunderstanding of what I was required to do after my first responses and submission of reviewers’ comment on 26th August 2024 because all the issues raised by the reviewers were minor. I just met a senior colleague today who identified the possible miscommunicate issue. After my first submission of reviewers’ comment, I assumed that we were done with the reviewer’s stage and moving on to the Editorial comment. I used the clean copy post reviewers’ stage to respond to the subsequent conversations. This was what gave birth to my responses quoted in the rejection decision and all other observations pointed out. Perhaps, I should have left the track changes made initially till the end. 

I will appreciate if this rejection decision can be reverted as this was my first time of encountering issues like this.

I have now uploaded manuscript with track changes in this resubmission. Also, I have included my responses to the editorial comments below

Editorial comments

RESPONSE: Done

2. To comply with PLOS ONE submissions requirements, in your Methods section, please provide additional information regarding the experiments involving animals and ensure you have included details on (1) methods of anesthesia and/or analgesia, and (2) efforts to alleviate suffering.

RESPONSE: It is now included. Line 133

3. In your Methods section, please provide additional details regarding participant consent from the owners of the animals. In the ethics statement in the Methods and online submission information, please ensure that you have specified (1) whether consent was informed and (2) what type you obtained (for instance, written or verbal). If the need for consent was waived by the ethics committee, please include this information.

RESPONSE: This is now included. Lines 99-100

Please upload the completed Content Permission Form or other proof of granted permissions as an """"Other"""" file with your submission.

RESPONSE: The map was self-constructed using baselayer from DIVA-GIS. Reference is now included in the figure description. Line 126-128

RESPONSES TO EDITORIAL’S COMMENTS ON 18TH SEPTEMBER 2024

1. To comply with PLOS ONE submissions requirements, in your Methods section, please provide additional information regarding the experiments involving animals and ensure you have included details on methods of anesthesia and/or analgesia, and efforts to alleviate suffering

RESPONSE: Please note that the manuscript reported a cross-sectional study and not an experimental study. Samples were taken once and death of an animal is not an outcome. However, I have highlighted in the track changes that minimal to no pain was ensured with application of methylated spirit for disinfection

---

## [Decision Letter · Decision Letter 2]

16 Dec 2024

GENETIC TRACKING OF CRIMEAN-CONGO HAEMORRHAGIC ORTHONAIROVIRUS IN HYALOMMA POPULATION INFESTING CATTLE IN NIGERIA

PONE-D-24-16845R2

Dear Dr. DAODU,

We’re pleased to inform you that your manuscript has been judged scientifically suitable for publication and will be formally accepted for publication once it meets all outstanding technical requirements.

Kind regards,

Gábor Kemenesi, Ph.D.

Academic Editor

PLOS ONE

Additional Editor Comments (optional):

The authors describe original and novel data from Nigeria, regarding the circulation of Orthonairovirus haemorrhagiae (CCHFV). As I cheched the history of this manuscript and evaluated the novel verison I found that authors addressed all key comments of the two previous reviewers. I would like to thank the work of the authors and reviewers which I believe resulted in a valuable piece to better understand the genomic diversity of CCHFV in Africa.

Reviewers' comments:

Reviewer's Responses to Questions

**Comments to the Author**

1. If the authors have adequately addressed your comments raised in a previous round of review and you feel that this manuscript is now acceptable for publication, you may indicate that here to bypass the “Comments to the Author” section, enter your conflict of interest statement in the “Confidential to Editor” section, and submit your "Accept" recommendation.

Reviewer #2: All comments have been addressed

2. Is the manuscript technically sound, and do the data support the conclusions?

Reviewer #2: Yes

3. Has the statistical analysis been performed appropriately and rigorously? 

Reviewer #2: Yes

4. Have the authors made all data underlying the findings in their manuscript fully available?

Reviewer #2: Yes

5. Is the manuscript presented in an intelligible fashion and written in standard English?

Reviewer #2: Yes

6. Review Comments to the Author

Reviewer #2: (No Response)

7. PLOS authors have the option to publish the peer review history of their article (what does this mean?). If published, this will include your full peer review and any attached files.

Reviewer #2: No

---

## [Editor Report · Acceptance letter]

14 Jan 2025

PONE-D-24-16845R2 

PLOS ONE

Dear Dr. Daodu, 

I'm pleased to inform you that your manuscript has been deemed suitable for publication in PLOS ONE. Congratulations! Your manuscript is now being handed over to our production team.

Kind regards, 

on behalf of

Dr. Gábor Kemenesi 

Academic Editor

PLOS ONE